# Forming Features and Properties of Titanium Alloy Billets After Radial-Shear Rolling

**DOI:** 10.3390/ma12193179

**Published:** 2019-09-27

**Authors:** Mikhail Mikhailovich Skripalenko, Sergei Pavlovich Galkin, Boris Vladimirovich Karpov, Boris Alekseevich Romantsev, Lyudmila Mikhailovna Kaputkina, Andrei Vladimirovich Danilin, Mikhail Nikolaevich Skripalenko, Pavel Vladimirovich Patrin

**Affiliations:** 1Department of Metal Forming, National University of Science and Technology “MISiS”, Leninksi prospekt, 4, 119049 Moscow, Russia; glk-omd@yandex.ru (S.P.G.); boralr@yandex.ru (B.A.R.); poinson@inbox.ru (L.M.K.); danilinav@yandex.ru (A.V.D.); tfsmn@yandex.ru (M.N.S.); patrinder@yandex.ru (P.V.P.); 2Nauchno-proizvodstvennyj centr Obrabotka metallov davleniem Intitutskiy Proezd, 2, Zavoda Mosrentgen, 142771 Moscow Oblast, Russia; prokat21@mail.ru

**Keywords:** radial-shear rolling, three-high rolling mill, hardness, strength, titanium alloy, computer simulation, strain effective, stress triaxiality

## Abstract

Radial-shear rolling (RSR) of titanium alloy billets was realized in a three-high rolling mill. Experimental rolling was simulated using DEFORM software. The purpose was to reveal how stress-strain state parameters, grain structure and hardness vary along the billet’s radius in the stationary stage of the RSR process. It was also the goal to establish a relation between stress state parameters, hardness and grain structure. Changes in the accumulated strain and the stress triaxiality were established by computer simulation. Hardness and grain size changes were obtained after experimental rolling. The novelty aspect is that both computer simulation and experimental rolling showed that there is a ring-shape area with lowered strength in the billet’s cross-section. The radius of the ring-shape area was predicted as a result of the research.

## 1. Introduction

Modern classification of metal forming processes introduces radial-shear rolling (RSR) as screw rolling at high values (15–18° or more) of the rolls’ feed angle of the rolls [1]. Advantages and theoretical background of RSR are provided in detail in [2], and the forming features are given in [1,2]. Radial shear rolling mini-mills [2] are relatively widespread. These mini-mills are explored in Russia [2] for producing round bars from different materials with unique features. The results of investigation of RSR in mini-mills operating in Germany are given in [3]. The study represents a mathematical model of the process in details. Using an RSR mill located in South Korea, the authors of [4] concluded that RSR is a promising technique for producing gradient structures. The RSR mill 14/40 of Czestochowa University of Technology (Poland) was used in [5] to demonstrate that round bars, produced by RSR, have smaller ovality, and they are more straight as compared to the ones produced by longitudinal rolling. Using of the RSR mill 14/40 also shows that the RSR process can be run with high elongation factors in a single pass [6]. It makes the skew rolling more economical and reduces the losses for roll mechanical working [6].

RSR processes have unique features, among which are: the creation of locally expanding metal flow tubes in an integrally narrowing deformation region; the deceleration of inner layers of the billet flow with a simultaneous acceleration of outer layers flow; and the production of volume macro shear forming [2,3]. 

Unique features of the metal flow and the stress-strain state (SSS) during RSR change billet metal properties and the grain structure. The intensive shear strain created by three-roll RSR mini-mills significantly decreases the grain size of different materials such as steels [7], titanium alloys [8,9,10,11], magnesium and its alloys [4,5,6], copper and copper alloys [12,13,14], aluminum alloys [15].The fine-grain structure increases the range of round bar properties [2]; above all, plasticity and associated properties [5,8]. A simultaneous increase of plasticity and strength was obtained by RSR in [2].

The estimation of SSS at RSR is currently realized using a finite element method (FEM) of computer simulation. Piercing by the Diesher mill (a two-roll screw rolling mill with guiding discs) was investigated in [16] with the help of computer simulation that showed a difference in SSS for two- and three-high screw piercing [17]. The research of stationary and non-stationary stages of three-high RSR using the DEFORM software was carried out in [18]. It is worth notifying that SSS at screw rolling can be estimated experimentally by using laminated billets [19], but such a technique was applied only for two-high screw rolling rather than three-high. Only one of seven billets after rolling were suitable for investigation. Other billets collapsed while rolling. The techniques used in [19] require sufficient time for billets preparation and data processing.

The joint use of computer simulation and experimental estimation is an effective technique for the investigation of RSR as shown in [20]. A comparative analysis of computer simulation results for SSS and the microstructure formation at RSR is of researchers’ interest.

The objective of our research was to investigate the relationship between the forming features and the properties formation during RSR of titanium alloy billets. It was also the purpose of the study to define the type of fracture that may occur while RSR of titanium alloy billets. 

## 2. Materials and Methods 

RSR was carried out using a three-high MISIS-100T mill (The Electrostal Heavy Engineering Works JSC ("EZTM" JSC), Electrostal city, Moscow Region, Russian Federation) (Figure 1). The rolls feed angle β was 20 degrees, the inclination angle α was 10 degrees. The materials of the billets were Ti-6Al-4V titanium alloy. The reduction of the diameter during RSR was according to the route: 76 mm→65 mm→55 mm→48 mm, so the reduction of the diameter for each pass was 14%, 15%, 13%. The billets were preliminary heated till 940–950 °C for 1 hour. The phase transition for alpha+beta→beta occurs at 960–980 °C [21], if the temperature is higher, the roughening of the structure occurs, and the following heat treatment cannot change it [22]. When the temperature is significantly lower than the polymorphic transformation temperature, it is hard to have plastic forming because of high strain resistance. The heating length of 1 hour before RSR provides a uniform temperature for the whole billet’s volume. The heating till rolling temperature for 5–10 min was done between the passes. The angular velocity of the rolls equaled 9.42 rad^−1^. The round bars were air-cooled after rolling. The semicircular specimen was cut out in the middle of the bar to measure its hardness and microstructure along the cross section. The scheme for measuring hardness is presented in Figure 2. Each point represents the place where hardness was measured. The given hardness values are mean of the three measures from the area of 1 mm^2^. The distance between the points for measuring hardness was 4 mm both vertically and horizontally. The distance from the bar’s center was also calculated for each point of Figure 2. Etching was done using a mix of 10% HF and 30% HNO_3_ with an H_2_O basis. The photographs of the microstructure were obtained by means of Axio Scope A1 Carl Zeiss *Carl Zeiss* Microscopy GmbH, Jena, Germany). The grain size was estimated using a linear intercept method. Hardness (HV scale, 49 N load) was measured by Micromet 5101 (Buehler, Lake Bluff, Illinois, IL, USA).

SolidWorks 2009 software (Dassault Systemes, Waltham, Massachusetts, MA, USA) was used for the creation of 3D models consisting of rolls, billet, guiding tools, and a pusher. The models were saved in a “.stl” format and downloaded in the DEFORM pre-processor (Figure 3a), initial and boundary conditions were set in concordance with the experimental rolling (billet’s temperature, rolls’ angular velocity, feed and inclination angles values, etc.). The friction factor for the rolls was set 1 in the Siebel friction law. The friction factor for the guides was set equal 0.12, and for the pusher, it was 0.3. “Separable” option was also chosen in the “Inter-Object Data Definition” window of the DEFORM pre-processor to govern contact between the tools and the billet. These values were set according to the experience of the computer simulation of titanium billets RSR using DEFORM [18] and the recommendations for the simulation of titanium billets RSR provided by the official representative of SFTC (DEFORM software designer) [23] in Russia—TESIS company [24]. The rolls, the pusher, and the guides were taken to be rigid bodies, i.e., they were subjected to neither elastic nor plastic deformation during the simulation. The workpiece (titanium billet) was considered as a plastic object. The material for the billet was set as “Ti-6Al-4V [70-1860F(20-1000C)]” from the DEFORM materials library, the flow stress model was chosen $\bar{\sigma }=\bar{\sigma }\left(\bar{\varepsilon },\;\dot{\bar{\varepsilon }},\;T\right)$, where $\bar{\sigma }$ is the flow stress, $\bar{\varepsilon }$ is the strain, $\dot{\bar{\varepsilon }}$ is the strain rate, and T is the temperature. Other materials settings were set as default. The tetrahedral mesh was used for the billet with 70,000 elements (Figure 3b), the mesh type was set as “Relative”, the size ratio equaled 2. The mesh refinement in the contact area (between the rolls and the billet) was done by the DEFORM (Scientific Forming Technologies Corporation (SFTC), Columbus, Ohio, OH, USA) simulator (default option, provided by the available version of DEFORM). 

## 3. Results

Figure 4 shows how the bar microstructure varies along the radius in the cross section. The alpha phase is bright, the beta phase is dark. The microstructure was estimated for the points that lie on the radius with the 4 mm step. The photograph that corresponds to the closest to the bar surface area (Figure 4e) shows the microstructure with the grain size that is visually less than for the other areas. Figure 4f proves it quantitatively: the minimum grain size value is 6.3 µm for the alpha phase and 0.4 µm for the beta phase. At that, there is an increase of the grain size values followed by a decrease while moving along the bar’s radius. It means that the deformation near the bar surface in the cross section is higher than that of the other points on the radius. This feature of the deformation distribution was revealed after the DEFORM simulation. Seven points were chosen along the bar radius using the “point tracking” function of the DEFORM post-processor (Figure 5) after each of the three passes of RSR. The strain rate effective values were calculated for these points during all three passes. A change of the strain rate effective was integrated by time and the accumulated strain values were calculated for each point (Figure 6).

The points located at 16, 20, and 21 mm from the bar center accumulated the strain values with up to 7.5 times higher than those for the points located at 4, 8, and 12 mm from the bar center. Specified distribution of the accumulated strain is the most probable reason to form the grain structure shown in Figure 4: the smallest grain size for both phases, especially the beta phase, is near the bar’s surface (due to the maximum values of the accumulated strain near the bar’s surface).The data of Figure 4f shows that for the beta phase, the grain size near the surface is 2.1 times smaller, for the alpha phase, it is 1.2 times smaller. Figure 4 proves qualitatively the point provided by the inventors of the RSR process [1]. This point postulates that the deformation provided by the rolls does not completely penetrate the billet, and the axial zone is formed weaker than the others, and it may evaluate in the gradient structure [1,2,3]. The gradient structure after RSR was also detected in [4,9].

The distribution of the hardness values along the radius in the stationary stage was obtained as a result of the experimental research (Figure 7). The hardness values with respect to distance from the bar’s center were plotted for each point (see the points in Figure 2); after that, the trend line governed by quadratic equation was built. 

Hardness correlates with strength, mainly with the ultimate strength [25,26]. The lower the hardness, the lower the strength. According to Figure 7, the lowest strength (and the lowest hardness) is located approximately half of the billet radius. It is known that at three-high screw rolling there is a danger of the ring shaped fracture [20,27], and for that ring rigidity, the coefficient value under stress condition is the highest [1,20]. The rigidity coefficient under stress condition divided by three is known as stress triaxiality, which researchers recommend using for fracture prediction during screw rolling processes [28,29]. The mean values of the rigidity coefficient under stress condition were calculated for the points in Figure 5 using the DEFORM post-processor. A change of hardness along the bar radius and a change of the rigidity coefficient under stress condition along the bar radius are presented in Figure 8. The values of “mean stress” and “stress effective” were calculated for the points in Figure 5. The data was exported from the DEFORM post-processor and downloaded into Microsoft Excel. The “mean stress” values were divided by the “stress effective” values for each point (at each step of the simulation) to calculate the rigidity coefficient under the stress condition values. The average value of the rigidity coefficient under stress condition was calculated for each point. The trend line governed by quadratic equation was built. This trend line and the trend line from Figure 7 were placed in one diagram (Figure 8). 

The change of the rigidity coefficient under stress condition (Figure 8) shows that a fraction is mostly probable 8–9 mm form the bar center (0,03 value is reached, which is maximum); the hardness, which has minimal values of 11–12 mm, from the bar center. The shape of the curves in Figure 8 reveals correlation between the compared parameters, and the curves have common extremal character of changing along the bar’s radius. It is also worth notifying that the “extremal” character is demonstrated for the grain size changing (Figure 4): the local maximum for the alpha phase is for 10 mm from the center and for the beta phase, it is 9–10 mm from the center. A probable difference between the positions of extremums of the experimental graphs (grain size and hardness) and the simulation data graph (the rigidity coefficient under stress condition) is that the microstructure evolution model was not inserted in the DEFORM post-processor. To date, there is no research dedicated to computer simulation of microstructure evolution during screw rolling processes, however such computer simulation was done for other metal forming processes [30]. Even if such a computer model was designed, it would be time demanding because of the microstructure evolution model calibration that could take weeks. 

A tendency to a ring-shaped fracture during three-high screw rolling should appear in the “extremal” character of the stress-strain state parameters changing along the bar radius. Apart from having maximum at the rigidity coefficient under stress condition curve (Figure 8) or the inflection point at strain rate effective [9], Figure 6 demonstrates a sharp increase at 12 mm from the bar center and farther. The accumulated strain increase is within 20% between 0 and 12 mm from the bar’s center, but the increase reaches 330% from 12 mm till the bar’s surface. The results of [5,6,9,12] taken together reveal a relation between changing of the grain size, the hardness, and the strain effective in the cross section of the bar at the stationary stage of RSR. But these relations are not interpreted in terms of prediction and tendency to fracture. 

## 4. Conclusions

Radial-shear rolling of Ti-6Al-4V titanium alloy billets with a diameter reduction according to the route 76 mm→65 mm→55 mm→48 mm was done so the diameter reduction was 14%, 15% and 13% correspondingly. The experimental rolling was simulated using the DEFORM software. The research showed that:

1. The accumulated strain change along the bar radius in the stationary stage of RSR obtained by the DEFORM simulation demonstrates that near the bar’s surface layers have values that are significantly (up to 7.5 times) higher than those in the central layers. According to the experimental results, it leads to a smaller grain size near the surface as compared to the central part of the bar;

2. The change of hardness along the bar radius shows that there is a ring-shaped area that has the lowest strength and is mostly liable to fracture. The change of the grain size shows that the maximum values are reached at some distance from the center. The DEFORM simulation results provide a change of the rigidity coefficient under stress condition, or “stress triaxiality,” with maximum values and, hence, maximum liability to fracture between the bar center and the surface, which also yields a ring-shaped pattern. According to the experiments, the radius of the ring zone is 9–12 mm; whereas from computer simulation, the radius of the ring zone is 8–9 mm;

3. A tendency to a ring-shaped fracture may also be illustrated by the fact that an increase of the accumulated strain is not greater than 12 mm from the bar center and it does not exceed 20%, whereas between 12 mm and up to the bar surface, there is a sharp increase up to 330%.

## Figures and Tables

**Figure 1 materials-12-03179-f001:**
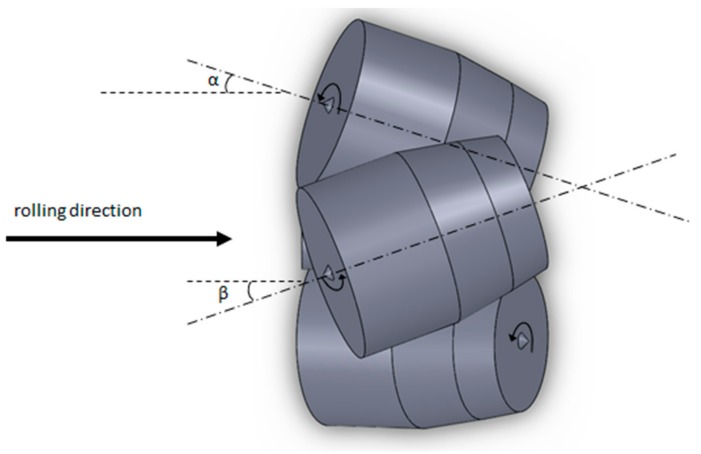
Scheme of radial shear rolling of the MISIS-100T rolling mill.

**Figure 2 materials-12-03179-f002:**
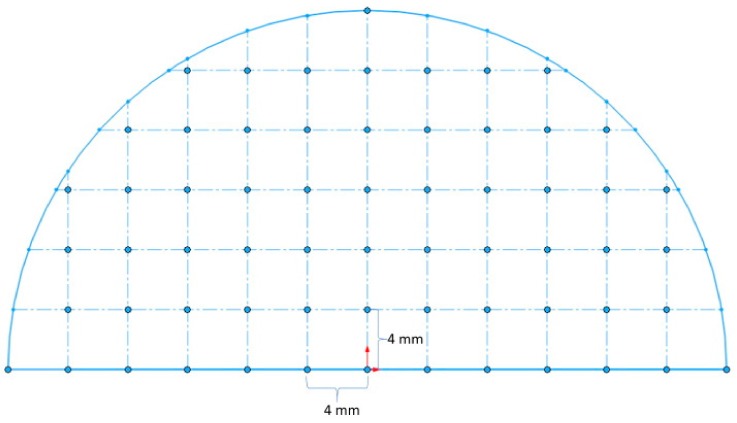
Points for measuring hardness in the round bar cross section.

**Figure 3 materials-12-03179-f003:**
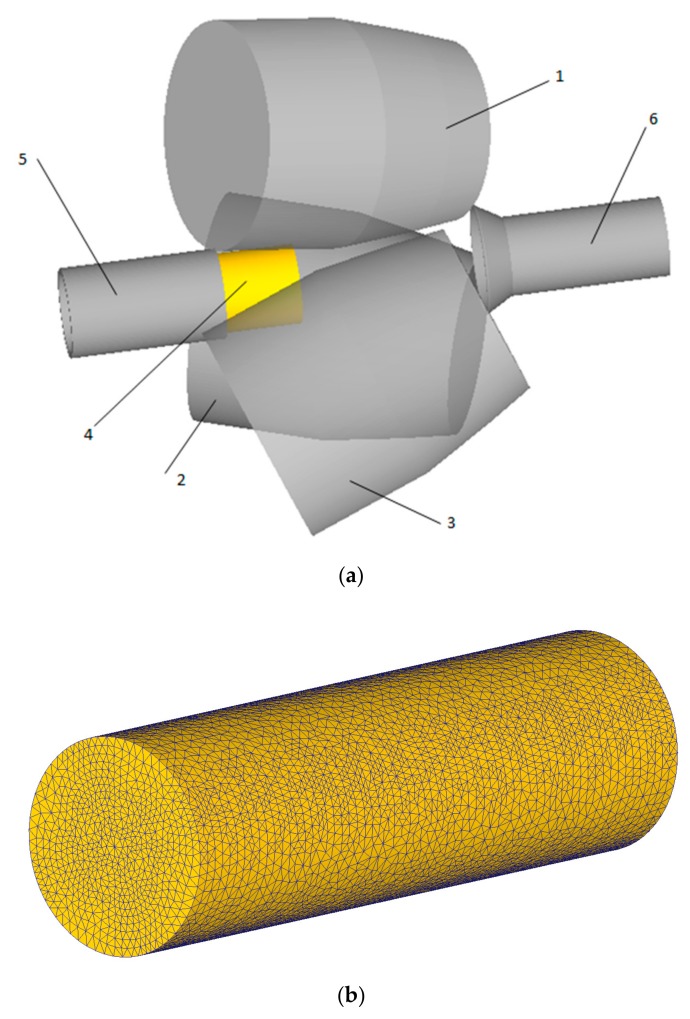
RSR model in DEFORM (**a**): 1,2,3 are the rolls, 4 is the billet, 5 is the entry side guide, 6 is the exit side guide and the billet with the tetrahedral mesh (**b**).

**Figure 4 materials-12-03179-f004:**
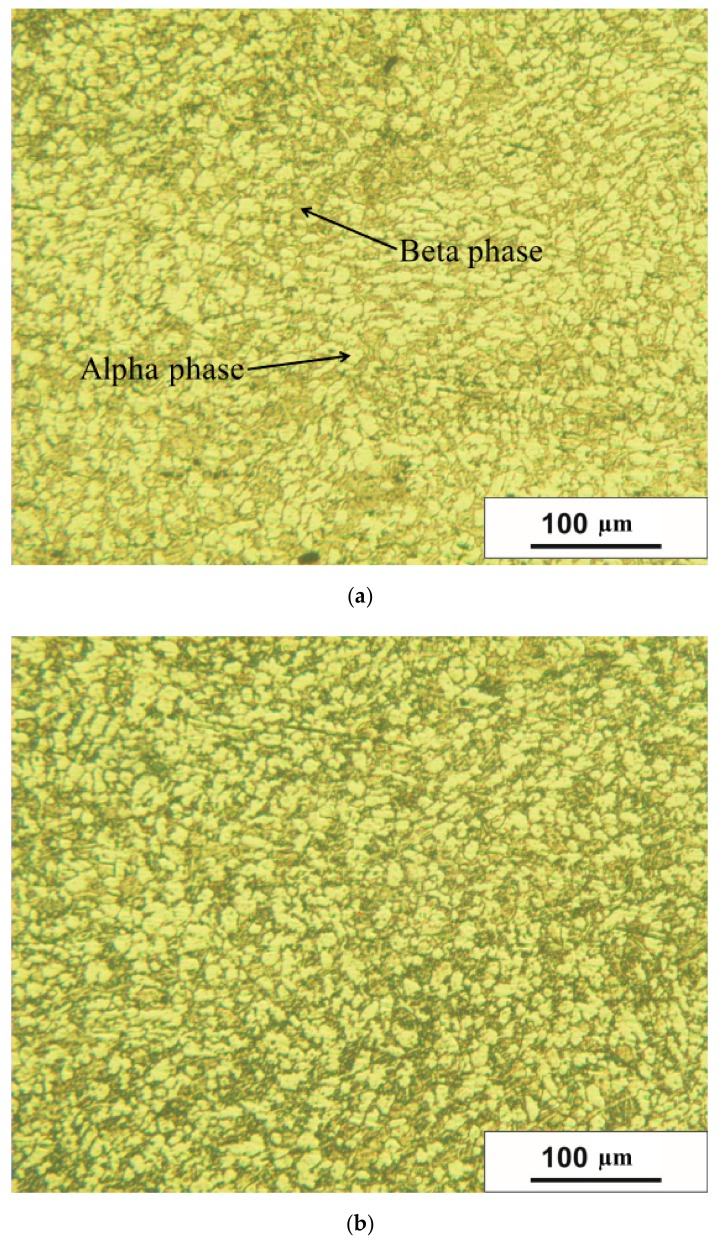
The bar microstructure in the cross section after RSR: (**a**) is the center, (**b**) is 5 mm from the center, (**c**) is 10 mm from the center, (**d**) is 15 mm from the center, (**e**) is 20 mm from the center, (**f**) is the grain size change along the bar’s radius, 1 is the alpha phase (bright), 2 is the beta phase (dark).

**Figure 5 materials-12-03179-f005:**
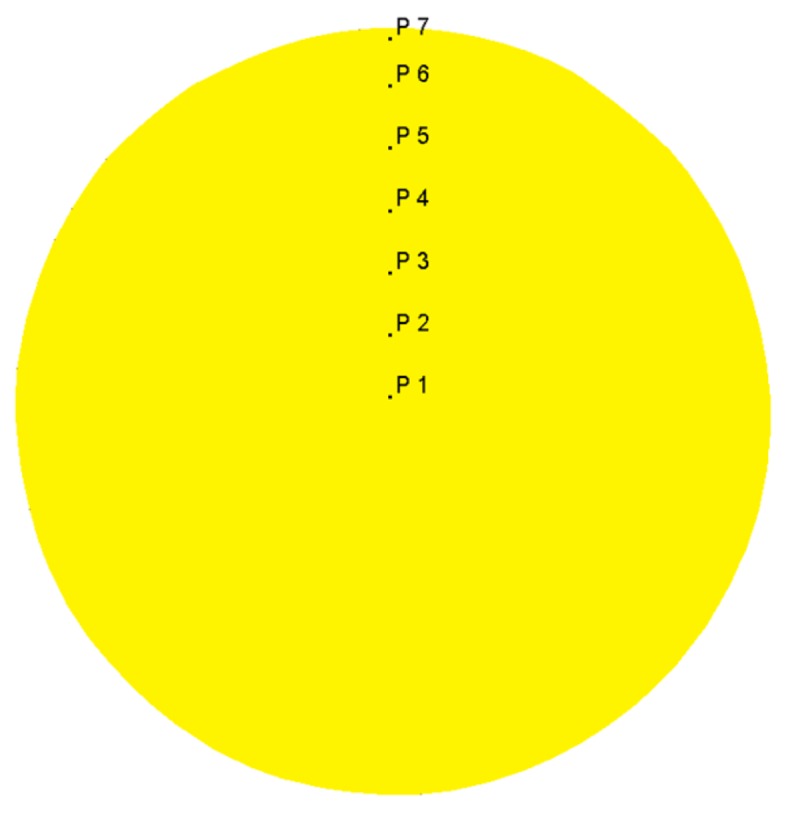
Points for calculating stress-strain parameters values using the DEFORM post-processor.

**Figure 6 materials-12-03179-f006:**
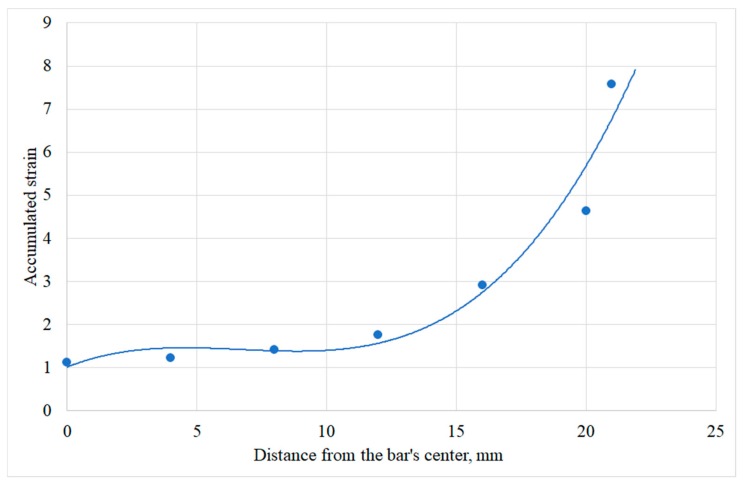
The change of the accumulated strain along the bar radius after 3 passes of RSR.

**Figure 7 materials-12-03179-f007:**
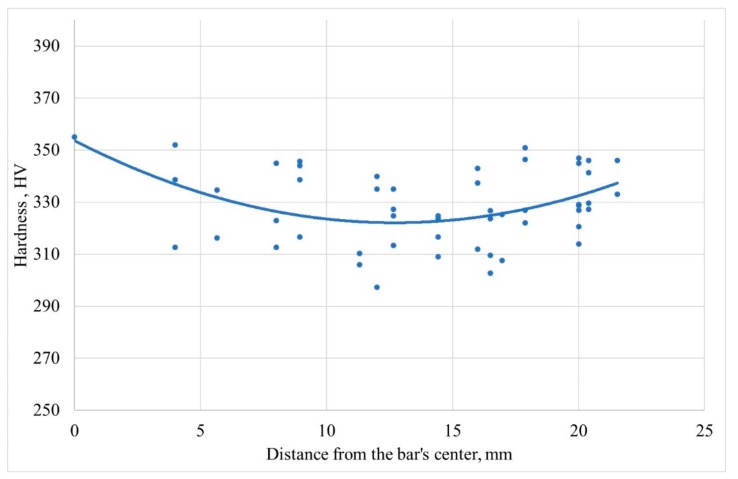
Change of hardness along the bar radius after 3 passes of RSR.

**Figure 8 materials-12-03179-f008:**
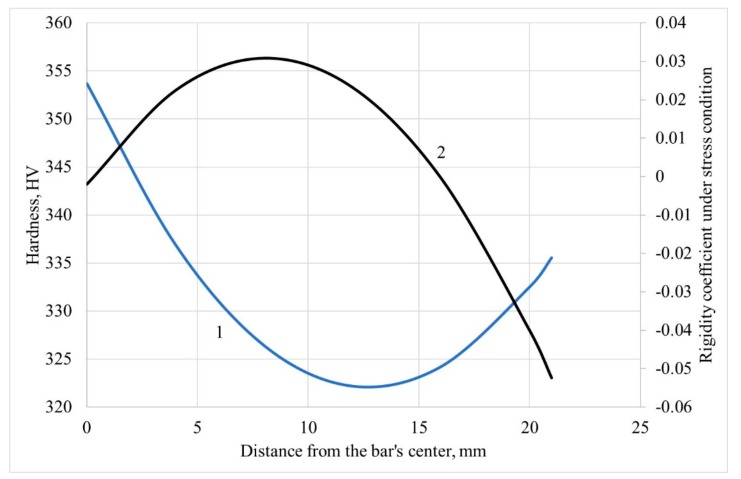
Change of hardness (1) and rigidity coefficient under stress condition (2) along the bar radius.

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
