# Peer review of "Forming Features and Properties of Titanium Alloy Billets After Radial-Shear Rolling"

_materials, 2019, doi:10.3390/ma12193179_

Round 1
Reviewer 1 Report
In this work, Radial-shear rolling of titanium alloy billets has been realized in a three-high rolling mill. Then the experimental rolling has been simulated using DEFORM software. Before further consideration the following issues should be considered and addressed:
The English of the manuscript should be double checked to correct all the grammatical and typos errors. The novelty aspect of the work is missing. This aspect should be highlighted. It is recommended to replace 6Al-4V titanium alloy with Ti-6Al-4V alloy. It is highly recommended to report the reduction of the diameter during RSR by percentage, instead of mm. In this way it would be clear that has the reduction per pass been constant or not. What is the motivation of preliminary preheating? Why these temperatures are selected? Metallography procedure is missing and should be included in detail. How the friction factors are selected? Why 1, 0.12 and 0.3 are selected? In figure 4, in the scale bar part, Mμ should be replaced by mμ. The self-citation is more than the standard limit and this aspect should be considered and addressed. 10 papers out of 27 are the papers of the second author.
Author Response
Point 1. The novelty aspect of the work is missing. This aspect should be highlighted.
Response 1. Information is added: lines 19-22. Newly added information is pasted green.
Point 2. It is recommended to replace 6Al-4V titanium alloy with Ti-6Al-4V alloy.
Response 2. It was replaced: line 60, line 183.
Point 3. It is highly recommended to report the reduction of the diameter during RSR by percentage, instead of mm. In this way it would be clear that has the reduction per pass been constant or not.
Response 3. Reduction is reported: line 62, line 183
Point 4. What is the motivation of preliminary preheating? Why these temperatures are selected?
Response 4. The reasons are given: lines 63-68, references 21 and 22 were added.
Point 5. Metallography procedure is missing and should be included in detail.
Response 5. Procedure was added: lines 73-75
Point 6. How the friction factors are selected? Why 1, 0.12 and 0.3 are selected?
Response 6. Explanation is provided in lines 85-88.
Point 7. In figure 4, in the scale bar part, Mμ should be replaced by mμ.
Response 7. It was replaced.
Point 8. The self-citation is more than the standard limit and this aspect should be considered and addressed. 10 papers out of 27 are the papers of the second author.
Response 8. We tried to reduce as much as we could. From 10 to 6. The truth is professor Galkin is one of the inventors of the RSR process and many researchers worldwide cooperate with him.
Reviewer 2 Report
This paper deals with a radial-shear rolling of a Ti-Al-4V alloy. Hardness measurements and microstructure observations in the middle of the tested specimen are performed. A simulation of the process is carried out with the DEFORM software.
This paper crucially lacks of information concerning experimental and modelling methods which impedes anyone to reproduce or compare the results. The apparatus used to perform the hardness measurements and microstructure observations are not specified. Setting parameters are not mentioned like maximal load (or penetration depth), beam power, etc. Moreover, the analysis of grain size variation is only based on eye observation whereas quantitative estimations of average grain size can easily be obtained by several methods (e.g. linear intercept method). Finally, about the model, nothing is said about the considered behavior of the titanium alloy (elastic? elasto-plastic? elasto-viscoplastic?), as well as the constitutive laws and material parameters that were used.
Author Response
Point 1. The apparatus used to perform the hardness measurements and microstructure observations are not specified. Setting parameters are not mentioned like maximal load (or penetration depth), beam power, etc.
Response 1. We added the information: lines 73-75.
Point 2. Moreover, the analysis of grain size variation is only based on eye observation whereas quantitative estimations of average grain size can easily be obtained by several methods (e.g. linear intercept method).
Response 2. We estimated grain size quantitatively: Figure 4f was added and discussion was given in lines 137-140.
Point 3. Finally, about the model, nothing is said about the considered behavior of the titanium alloy (elastic? elasto-plastic? elasto-viscoplastic?), as well as the constitutive laws and material parameters that were used.
Response 3. These details were added: lines 90-94
Reviewer 3 Report
In the abstract please clearly describe what the purpose of your study is. Please also indicate the findings.
In the introduction:
The novelty this paper brings should be expressed more explicitly. The objective is rather superficial. It sounds more like an engineering case study (application of method), rather than scientific work.
Radial shear rolling mini-mills [4-5] are relatively widespread, i.e., these mini-mills are 29 explored in Russia [6], Germany [3], South Korea [7], and Poland [8,9].
Please put more details about each research, their key findings/results etc. How those papers refer to your study.
The estimation of SSS at RSR is currently realized using a finite element method (FEM) of computer simulation. - any other methods excluding FEM?
Please clearly indicate how your method fit in the existing ones. What is new? What new information does your research bring to the field?
In materials and method:
Please clearly indicate governing equations for your FE model. Please indicate how meshing was done.
In results:
Please explain how the figure 4 was created.
Figure 4 shows how the bar microstructure varies along the radius in the cross section. The photo that corresponds to the most closed to the bar surface area (Figure 4e) shows the microstructure with the grain size that is visually less than for the other areas.
This should be quantified instead of "visually less".
There is hardly any discussion of this microstructure!
Hardness correlates with strength, mainly with the ultimate strength. The lower hardness is the lower strength is.
Please provide justification of this statement - Vickers hardness correlation with ultimate strength for your material... How is microstructure affecting this parameter?
It is known that at three-high screw rolling there is a danger
Justification needed!
Font size changed in this section?
How is the simulation model data validated? You present micro structure in Figure 4 so the experimental results were available. There is no reference
General: There are some language errors, formatting issues. Please use "." instead "," as decimal point.
Author Response
Point 1. In the abstract please clearly describe what the purpose of your study is. Please also indicate the findings.
Response 1: It was added: lines 15-16, lines 55-57. Findings are given in "Results" and "Conclusions". Added data is pasted green.
Point 2. The novelty this paper brings should be expressed more explicitly. The objective is rather superficial. It sounds more like an engineering case study (application of method), rather than scientific work.
Response 2. Please, see lines 19-22, lines 55-57,
Point 3. Radial shear rolling mini-mills [4-5] are relatively widespread, i.e., these mini-mills are 29 explored in Russia [6], Germany [3], South Korea [7], and Poland [8,9].Please put more details about each research, their key findings/results etc. How those papers refer to your study.
Response 3. We listed these references in order to show that RSR process is widespread.
Point 4. The estimation of SSS at RSR is currently realized using a finite element method (FEM) of computer simulation. - any other methods excluding FEM?
Response 4. We added the information you required: lines 47-51.
Point 5. Please clearly indicate how your method fit in the existing ones. What is new? What new information does your research bring to the field?
Response 5. It was added: lines 19-22, lines 178-181. We used both computer simulation and experimental rolling, it fits with specified in lines 52-53.
Point 6. Please clearly indicate governing equations for your FE model. Please indicate how meshing was done.
Response 6. Lines 92-94.
Point 7. Please explain how the figure 4 was created. Figure 4 shows how the bar microstructure varies along the radius in the cross section. The photo that corresponds to the most closed to the bar surface area (Figure 4e) shows the microstructure with the grain size that is visually less than for the other areas. This should be quantified instead of "visually less".
Response 7. The data was completed: lines 101-103, lines 105-107, lines 128-129, Figure 4f.
Point 8. There is hardly any discussion of this microstructure!
Response 8. We added discussion: lines 137-140.
Point 9. Hardness correlates with strength, mainly with the ultimate strength. The lower hardness is the lower strength is. Please provide justification of this statement - Vickers hardness correlation with ultimate strength for your material... How is microstructure affecting this parameter?
Response 9. References 25 and 26 were included. Hardness and grain size change both have "extremal" character of changing: lines 163-168, lines 191-197.
Point 10. It is known that at three-high screw rolling there is a danger. Justification needed!
Response 10. References 20 and 26 (line 149) are given.
Point 11. Font size changed in this section?
Response 11. Yes, it was. Now it is correct.
Point 12. How is the simulation model data validated? You present microstructure in Figure 4, so the experimental results were available. There is no reference
Response 12. Figure 8 is validation. Lines 162-167 are about comparing experimental and simulation results.
Point 13. General: There are some language errors, formatting issues. Please use "." instead "," as decimal point.
Response 13. Corrections were done for line 69 and for Figure 8
Round 2
Reviewer 1 Report
The revision is satisfactory and therefore the paper can be accepted.
Author Response
Reviewer: "The revision is satisfactory and therefore the paper can be accepted".
Reviewer 2 Report
The authors adequately considered the issues raised by the reviewers.
Author Response
Reviewer: "The authors adequately considered the issues raised by the reviewers."
Reviewer 3 Report
There is something wrong with this sentence:
The purpose was revealing the relationship between the ways stress-strain state parameters, grain structure and hardness vary along the billet’s radius in the stationary stage of the RSR process.
Radial shear rolling mini-mills [2] are relatively widespread, i.e., these mini-mills are explored in Russia [2], Germany [3], South Korea [4], and Poland [5,6].
Please do provide more information about those references. What were the findings of each of those research. At least one sentence per each item.
It would be nice to see the mesh for FEM. Any mesh refinement in the contact area. How was the contact between each parts handled?
the smallest grain size for both phases, especially beta phase, is near the bar’s surface (due to the maximum values of the accumulated strain near the bar’s surface).The data of Figure 4f shows that for beta phase grain size near the surface is 2,1 times smaller, for alpha phase – 1,2 times smaller.
This also requires some discussion!
The distribution of the hardness values along the radius in the stationary stage was obtained as 141 a result of the experimental research (Figure 7).
How was it obtained. Please provide the details so any other researcher can potentially reproduce your experiment.
Figure 8 comes from FEM analysis? Please explain how it was created.
Generally, English requires some extensive editing!
Author Response
Point 1. There is something wrong with this sentence:
The purpose was revealing the relationship between the ways stress-strain state parameters, grain structure and hardness vary along the billet’s radius in the stationary stage of the RSR process.
Response 1. The sentence was divided into two for better understanding (lines 15-18). All corrections for 2-nd revision are pasted blue.
Point 2. Radial shear rolling mini-mills [2] are relatively widespread, i.e., these mini-mills are explored in Russia [2], Germany [3], South Korea [4], and Poland [5,6].
Please do provide more information about those references. What were the findings of each of those research. At least one sentence per each item.
Response 2. Sentences were added for each reference with listing their findings (lines 31-40)
Point 3. It would be nice to see the mesh for FEM. Any mesh refinement in the contact area. How was the contact between each parts handled?
Response 3. Billet's mesh was added (Figure 3b, line 113). Sentence concerning mesh refinement was included (lines 105-107). The was contact for parts of the model was governed is described in lines 95-97.
Point 4. the smallest grain size for both phases, especially beta phase, is near the bar’s surface (due to the maximum values of the accumulated strain near the bar’s surface).The data of Figure 4f shows that for beta phase grain size near the surface is 2,1 times smaller, for alpha phase – 1,2 times smaller.
This also requires some discussion!
Response 4. Discussion was added (lines 156-159).
Point 5. The distribution of the hardness values along the radius in the stationary stage was obtained as 141 a result of the experimental research (Figure 7).
How was it obtained. Please provide the details so any other researcher can potentially reproduce your experiment.
Response 5. Details were provided (in order to reproduce experiment): lines 77-82, lines 162-164.
Point 6. Figure 8 comes from FEM analysis? Please explain how it was created.
Response 6. Figure 8 comes from FEM simulation and following data processing. Explanation was added: lines 176-182.
Point 7. Generally, English requires some extensive editing!
Response 7. Editing is about to be done by our university English editing service (native English editor).